# Additives for Efficient Biodegradable Antifouling Paints

**DOI:** 10.3390/ijms20020361

**Published:** 2019-01-16

**Authors:** Fabienne Faÿ, Maëlle Gouessan, Isabelle Linossier, Karine Réhel

**Affiliations:** Université Bretagne Sud, EA 3884, LBCM, IUEM, F-56100 Lorient, France; maelle.gouessan@univ-ubs.fr (M.G.); isabelle.linossier@univ-ubs.fr (I.L.); karine.rehel@univ-ubs.fr (K.R.)

**Keywords:** antifouling, copper paint, additives, biofilm

## Abstract

The evolution of regulations concerning biocidal products aims to increase protection of the environment (e.g., EU Regulation No 528/2012) and requires the development of new non-toxic anti-fouling (AF) systems. The development of these formulations implies the use of ingredients (polymers, active substances, additives) that are devoid of toxicity towards marine environments. In this context, the use of erodable antifouling paints based on biodegradable polymer and authorized biocides responds to this problem. However, the efficiency of paints could be improved by the use of specific additives. For this purpose, three additives acting as surface modifiers were studied (Tween 80, Span 85 and PEG-silane). Their effects on parameters involved in antifouling efficiency as hydrophobicity, hydration and copper release were studied. Results showed that the addition of 3% of additives modulated hydrophobicity and hydration without an increase of copper release and significantly reduced microfouling development. Efficient paints based on biodegradable polymer and with no organic biocide could be obtained by mixing copper thiocyanate and additives.

## 1. Introduction

Biofouling can be defined as the accumulation of micro- and macro-organisms on artificial surfaces immersed in seawater. To control biofouling, antifouling (AF) paints have been developed and commonly used for several decades. They contain polymers as binders, toxic compounds, called biocides, which are leached from the paint matrix, and additives (thixotropic agents, pigments, viscosity modifiers). Biocides are based on copper compounds (copper oxide or copper thiocyanate) associated with booster biocides. These organic biocides are intended to be environmentally less harmful than the organotin biocides used in the 1970s. However, alternative strategies are researched because problems of toxicity for marine species and an accumulation of substances in seawater persist [1,2]. Among them, the use of natural antifouling compounds has received a lot of attention. For example, papain, butenolide or cardenolides recently showed an interesting efficiency against biofouling [3,4,5]. However, several restrictions limit the use of these products: large scale production, degradation, compatibility with paint matrix, release characteristics and costs are the main difficulties impeding their development [6]. Moreover, recent regulation concerning the use of biocides (EU Regulation No 528/2012) known as Biocide Product Regulation (BPR) also restricts their industrial development for reasons concerning costs. Actually, only a few biocides are authorized by European Union and commercialized: 3 copper derivatives (copper, copper thiocyanate and dicopper oxide), 5 booster biocides (DCOIT, Zineb, copper pyrithione, zinc pyrithione and a substituent of copper called Tralopyril). Currently, antifouling paint researchers have to look for a compromise between efficiency of coatings and their impact on the environment.

The aim of this work is to study paints based on a copper derivative such as copper thiocyanate, but devoid of booster biocides. Copper is an effective biocide against algae and mollusks. Moreover, a lower amount of copper thiocyanate is needed than copper oxide for the same level of efficiency [7]. However, to improve its efficiency and to enlarge the activity spectrum of paints, an additive acting as surface modifier could be added. Its role is not to produce a biocidal effect but instead the promotion of an anti-adhesive effect by modifying wettability of surface and paint surface/organisms interactions. Fouling release (FR) coatings based on poly(dimethylsiloxane) (PDMS) rely on this principle by decreasing surface energy [8]. However, several publications have mentioned the combination of antifouling and fouling release concepts to develop new hybrid materials effective in marine antifouling protection. Azemar and al. have proposed a hybrid system based on triblock copolymer poly(ε-caprolactone)-block-poly(dimethylsiloxane)-block-poly(ε-caprolactone) to mix the properties of erosion/biocide release used in antifouling systems and hydrophobicity properties through the use of PDMS [9]. Afterwards, Yang et al. have confirmed the need to combine the concepts of “antifouling” and “fouling release” [10].

The surface wettability plays a major role in antifouling performance [10]. It can be modulated by the use of surfactants. For example, Tween 20 has improved the antifouling characteristics of membranes by adsorption at interfaces [11,12]. In antifouling applications, we have previously shown that Tween 85 disturbed interactions between colonizing organisms and surfaces by decreasing their hydrophobicity and thus a physical repelling of Tween 85 has been hypothesized [13]. Another strategy concerns the use of grafted surfaces. Surface-grafted poly(ethylene glycol) (PEG) molecules are known to prevent protein adsorption and coatings based on PDMS-g-PEG have been studied in seawater [8,14,15]. In this study, bacteria and diatoms adhesion were inhibited. Recently, Jimenez-Pardo et al. have proposed hydrophilic self-replenishing coatings based on polycarbonate-poly(ethylene glycol) methyl ether polyurethane exhibiting low proteins adhesion values [16].

Hence, this work has studied the effect of three additives incorporated in a copper paint: Tween 80 and Span 85, two no ionic, hydrophilic and hydrophobic surfactant respectively and a PEG-silane. The last one is considered as a surface modifier because of its properties such as hydrophilicity, flexibility, high exclusion volume in water and non toxicity [17]. The paint had an additive free formulation. Parameters as surface hydrophobicity, paint hydration and copper release were evaluated. The anti-microfouling and anti-macrofouling efficiencies of paints were studied as well as their toxicity.

## 2. Results and Discussion

### 2.1. Effect of Additifs on Paints Characteristics

To study the effects of additives on hydrophobicity, hydration, and copper release, four paints were studied: three paints containing a surface modifier at 3% (*w*/*w*) (Figure 1) and one paint without additives. The same formulation was used for all the paints. Only the additive nature changed. Tween 80 is a hydrophilic surfactant (hydrophilic lipophilic balance (HLB 16.7), whereas Span 85 is a hydrophobic one (HLB 1.8). PEG-silane is a biocompatible polymer. Immobilized onto surfaces, it confers protein and cell resistance [18]. Here, it was used as a surface modifier (wettability and steric hindrance). Calcium carbonate was used to complete the formulation. It is a neutral charge often incorporated in antifouling paint.

#### 2.1.1. Paint Surface Hydrophobicity

Water contact angles were measured to evaluate changes in hydrophobicity of coating surfaces. Decrease in water contact angle was attributed to coatings with higher wettability, whereas an increase reflected a more hydrophobic surface. Results are shown in Figure 2.

Water contact angle measurements were realized before and after 14 and 28 days of immersion in distilled water. Before immersion, the paint without additive showed a hydrophobic behavior with a contact angle of 90.2 ± 0.6°. The addition of the hydrophilic surfactant (Tween 80) induced a significant reduction of contact angle (79.9 ± 2.3°) (*p* < 0.05) whereas the incorporation of Span 85 and PEG-silane significantly increased hydrophobicity (121.5 ± 4.2° and 102.5 ± 1.9° respectively, *p* < 0.05). During immersion, the paints behaviors were different. No significant evolution was observed for both surfactants (*p* > 0.05). Conversely, a significant (*p* < 0.01) continuous decrease of contact angle was observed for PEG-silane: the surfaces seemed less permeable. A migration of the PEG chains at the surface of the coatings was presumed during immersion. For paint without additive, a significant decrease (*p* < 0.01) of water contact angle was observed after 14 days, then the hydrophobicity remained stable. Similar results have already been observed previously for paint based on Poly(ε-caprolactone) homopolymer. The decrease of water contact angle during the first days of immersion was explained by polymer hydration and degradation processes [9].

#### 2.1.2. Paint Hydration

The global hydration of paints was followed by Karl Fisher titration during immersion in distilled water (Figure 3). The additives led to a water absorption significantly increased compared to the paint without additive (1% average). Tween 80 and Span 85 showed a constant hydration rate (about 8% taking to account the standard deviations). More variations were observed for PEG-silane. A decrease of hydration was observed to the 14th day, then the rate increased to reach 8%.

#### 2.1.3. Copper Release

Additives included in the formulation could modulate copper release. Hence, the copper thiocyanate detected in surrounding water was quantified. Figure 4 shows cumulative release of copper during 30 days of immersion. As shown, copper thiocyanate was released faster from paint composed of PEG-silane than all other paints. PEG-silane improved copper release. The cumulative amount released after 28 days was found to be 3.05 µg/cm^2^, whereas only 0.58 and 0.06 µg/cm^2^ were quantified for the paints containing Tween 80 and Span 85, respectively. The paint without additive did not show significant difference of copper release, the copper cumulative release after 28 days was found to be 0.18 µg/cm^2^. These values corroborated with known data of copper release from erodible paints for which the amount of copper release can be relatively weak [19,20]. However, these release rate differences were not significant: whatever the additive, less of 1% of copper thiocyanate was released after 28 days of immersion.

Hence, the presence of additive did not increase the copper thiocyanate release to the current rates of commercial paints and thus limit the environmental impact.

### 2.2. Antifouling Properties

*In situ* immersions in natural seawater reveal the anti-microfouling and anti-macrofouling properties of paints. Paints were immersed for 9 weeks and 13 months (from April 2017 to May 2018) to evaluate the impact of additives on microalgal and macro-fouling development, respectively.

#### 2.2.1. Anti-microfouling Activity

Coatings were immersed for 9 weeks in natural seawater and then were observed by CLSM (Figure 5). The biofilm was quantified by COMSTAT analysis to obtain biomass and average thickness of microalgae on paints (Figure 6). Microalgae are a major colonizer of antifouling paint and several publications have mentioned their pertinence as model in antifouling research [21,22,23].

Paint without additive showed a denser and thicker biofilm than paints with additive at every observation time (Figure 6). At 4 weeks, a significant decrease (*p* < 0.01) of biovolume and average thicknesses of biofilm were observed for paints with additive comparatively to paint without additive. After this period, the nature of micro-organisms developed on paint without additive was noticeably different: the colonization step was more advanced with the presence of chains of microalgae. Hence, the quantification of biofilms on paint without additive became technically difficult: the quantification was underrated. Nevertheless, the values quantified on this paint were always higher than on the other paints. In the case of paint with Tween 80 as additive, a decrease of biomass and average thickness was observed after 7 weeks. This result can be explained by a mechanical erosion of paint surface, as already observed in the case of erodible paints [9,24].

Hence, the presence of 3% additive confirmed the reduction of microalgal biofilm development. This was particularly the case for both surfactants whatever their hydrophilic/hydrophobic balance.

#### 2.2.2. Antimacrofouling Activity

To test the antimacrofouling activity for a longer period, paints immersed in natural seawater in static conditions were visually observed during 13 months (Figure 7). To quantify the efficiency of coatings, an efficiency factor (*N*) was determined (Figure 8). All paints showed lower *N* values than an unprotected surface (*N* = 30 after 3 months of immersion), confirming the efficiency of paints: in contrast to the case for the unprotected surface, no adherent macrofoulers were observed. All paints were colonized continuously during the first months, however, the rate of colonization of paints varied. Paint without additive reaches a value of 10 after only 3 months, paints containing Tween 80 after 5 months and Span 85 and PEG-silane after 7 months. Their efficiency was then similar: *N* values were constant and identical (*N* = 10) from 7 months to 13 months of immersion. No effect of additives was observed for the long term. The efficiency (*N* is lower than the unprotected surface) could probably be attributed to the presence of copper thiocyanate.

### 2.3. Paint Toxicity Against Microfouling

Paints with additive showed effective efficiency against micro-organisms as microalgae. Hence, the paints toxicity was studied in vitro against marine bacteria and microalgae. Results were confronted with those obtained for a commercial paint containing copper and booster biocides.

#### 2.3.1. Evaluation of Anti-bacterial Activity

The inhibition zone for the three marine bacteria *Pseudoalteromonas* sp. 5M6, *Bacillus* sp. 4J6 and *Paracoccus* sp. 4M6 was evaluated in the presence of paints (Table 1). Although the paint without additive contained copper, no bacterial inhibition was observed. This result was consistent with previous results [22]. None of the paint with tested additive affected bacterial growth except for *Paracoccus* sp.: a low inhibition was observed in the presence of Tween 80 and PEG silane. Conversely, commercial paint showed antibacterial activity on the three strains, where an inhibition diameter more than 0.3 cm was observed. This antibacterial activity was probably due to the presence of booster biocides. Hence, paints based on additives were far less toxic than commercial paint.

#### 2.3.2. Evaluation of Anti-microalgal Activity

Table 2 shows the zone of inhibition for three microalgal strains in the presence of paints. As for anti-bacterial tests, paint without additive was not toxic against microalgae and commercial paint showed the highest toxicity level. For paints containing additive, the effects were different, depending on the microalgal strains. The highest effect was observed in the presence of Span 85. Moreover, they were higher for microalgal strains than for bacteria. However, in all the cases, paints based on copper thiocyanate and 3% additive were less toxic than the commercial paint.

### 2.4. Can Additives Improve the Efficiency of Erodible Paints?

Surface modifiers as additives can play an important role in antifouling efficiency. In fact, the most effective approach to developing new antifouling paints entails making a compromise between efficiency and toxicity. Paints formulated in this paper were based on biodegradable polymer [25], only one biocide, the copper thiocyanate (authorized by the EU Regulation) and no toxic additives. The impact of additives was principally observed in the first event of immersion. Hence, their presence in the formulation reduced the microfouling development, which was explained by a higher hydration of paints. Hydration allowed a reduction of organisms-surfaces interactions: the establishment of biofilm was disturbed. The higher hydration did not accelerate the copper release, which remained particularly low in all the cases. Hence, paints proposed in this paper were composed of eco-friendly ingredients with low toxicity that retained their efficiency.

## 3. Materials and Methods

### 3.1. Chemical Products

Tween 80 (Polyethylene glycol sorbitan monoleate) and Span 85 (Sorbitan trioleate) were purchased from Sigma-Aldrich (Saint Louis, MI, USA). PEG-silane (Silquest A1230, molecular weight = 546 g·mol^−1^) and ingredients of formulation (Solvents, plastifiant, calcium carbonate and fillers) were supplied from Momentive and Nautix Company respectively. The binder was a biodegradable polymer called poly(ε-caprolactone-co-δ-valerolactone) (P(CL-VL) 80-20) synthetized by Mäder (Lille, France) following the protocol described in Loriot et al. [25]. The molecular weight (Mn) of the polymer was 19,000 g·mol^−1^ with a polydispersity of 1.5.

### 3.2. Paints Formulation and Coupons Preparation

Binder was solubilized in solvent during 24 h (25 °C, 70 rpm) then paints were formulated by dispersing all ingredients (Table 3) under mechanical vigorous agitation (PBD40, Bosch) at 600 rpm. Once all ingredients added, the agitation was maintained for 15 min at 1000 rpm. Then the paints were filtered through a sifter (100 µm).

A layer of wet film (200 µm thick) was deposited with an automatic film applicator (ASTM D823 Sheen instrument) on a polycarbonate support. Then, the specimens were dried at 20 °C for one week.

A commercial paint called A4T was furnished by Nautix Compagny (Guidel, France).

### 3.3. Karl Fisher Titration

Paints plates were immersed in Artificial SeaWater (ASW, 33 g·L^−1^, Sigma Aldrich). Pieces of films (2 cm in diameter) were cut off in order to quantify the water amount present in films. The Karl-Fisher titration was performed with a Coulometer Methrom KF831 equipped with an Oven Methrom 860KF Thermoprep (150 °C) under an air flow of 60 mL·min^−1^. The reactant used was Hydranal-coulomat AG. The experiment was conducted in three triplicates for each sample.

### 3.4. Contact Angles Measurement

Measurements were obtained at room temperature with a contact angle Digidrop GBX (Dublin, Irland) equipped with a syringe, a video camera, and an acquisition of angle measurement. Five water droplets of 2 µL were measured at 0, 15 and 30 s after contact between the drop and the paint surface. The indicated values are an average of 5 measurements obtained on different areas of films.

### 3.5. Copper Thiocyanate Release

The cyclic voltammetric stripping (CVS) studies were carried out in determination mode on a software (Viva 2.0) connected to Metrohm 884 Professional VA. The voltammetry cell consists of a three electrodes assembly and a stirrer with hanging mercury drop electrode as a working electrode (Multi Mode Electrode pro, Metrohm; 6.0728.120 and 6.1246.1) a platinum wire (Metrohm; 6.0343.100) as auxiliary electrode leading the electric current to the working electrode and Ag/AgCl (satured KCl 3.0 M) electrode (Metrohm; 6.1204.50) as a reference electrode with a constant potential.

Analysis were carried out using the standard addition method. Thus 4 mL of sample solution were transferred into the electrolysis cell, containing 10 mL water HPLC grade (VWR) and 1 mL of electrolyte solution (21.6 g KCl, 50 mL NaOH at 30%, 28.4 mL of acetic acid and water up to 1 L with a pH = 4.6 ± 0.2). The solution was purged with pure nitrogen during 5 min. The accumulation potential was applied to a new mercury drop (5 mm) while the solution was stirred at 2000 rpm for 60 s. At the end of the accumulation period, the stirring was stopped and a 60 seconds rest period was allowed for the solution to become quiescent. Then the voltammogram was recorded by scanning the potential toward the positive direction over the range –0.9 to +0.2 V. Copper was detected around −0.1 V. The standard solution of Cu (VWR) at 2 mg·L^−1^ was prepared from standard solution at 1 g·L^−1^. The volume of the standard solution was 100 µL. All measurements were made at room temperature.

### 3.6. Anti-Bacterial Activity

The marine bacteria used (*Bacillus* sp. (4J6), *Pseudoalteromonas* sp. (5M6) and *Paraccous* sp. (4M6)) were grown on a Zobell medium: Artificial Seawater 30 g/L, Tryptone 4 g/L, Yeast Extract 1 g/L. Bacterial cultures were incubated at 10^6^ cfu/mL during 48 h under agitation. Planktonic cultures were maintained at 20 °C whilst shaking. These bacteria were used because they are pioneer adherents. Strains were isolated from the surface of a glass cover immersed in natural seawater (Morbihan gulf, France) for 6 h [26]. 5M6, a Gram negative bacteria, was affiliated to the *Pseudoalteromonas* genus. The Gram positive bacteria 4J6 clustered with the genus *Bacillus* (100% similarity) and 4M6 was affiliated to *Paracoccus* sp.

The zone of inhibition assay on solid media was used for determination of the antimicrobial effects of paints against *Bacillus* sp. (4J6), *Pseudoalteromonas* sp. (5M6) and *Paraccous* sp. (4M6). 10 mL of molten Zobell agar was inoculated by 1 mL of bacterial cultures (colony count of 1 × 10^7^ UFC/mL). Coupons of paints (2 cm diameter) were placed on the bacterial carpets and incubated at 20 °C for 48 h in an appropriate incubation chamber. The plates were examined, and the diameter of the inhibition zone was measured (in centimeters). These experiments were repeated three times for each sample.

### 3.7. Anti-Microalgal Activity

Three marine strains *Cylindrotheca closterium* (Diatomophyceae, AC515), *Porphyridium purpureum* (AC122) and *Exanthemacrysis gayraliae* (AC15) were used. Microalgae were obtained from the Culture Collection of Algae of the University of Caen (France). Microalgae were grown in an ASW-based culture medium with Guillard’s F/2 Marine Enrichment Basal Salt Mixture (Sigma Aldrich, Saint Louis, MO, USA), in sterile conditions at 20 °C. Guillard’s F/2 was added after sterilization and the culture medium was stored at 4 °C before use.

The zone of inhibition assay on solid media was used. 10 mL of molten medium agar was inoculated by 1 mL of microalgal cultures (1 × 10^5^ cells/mL). Coupons of paints (1.5 cm diameter) were placed on the microalgal carpets and incubated at 20 °C for five days in phytotronic chambers (controlled illumination of 150 µmol. photons.m^−2^·s^−1^ white fluorescent lamps with a 16h:8h light:dark cycle). The plates were examined, and the diameter of the inhibition zone was measured (in centimeters). These experiments were repeated three times for each sample.

### 3.8. Anti-Microfouling Properties

Paints (2 × 5 cm) were exposed in natural seawater, at a depth of 50 cm (Atlantic Ocean, W 47°43’8.39” N 3°22’7.38”, Larmor Plage, France). The study began in April 2017. The seawater characteristics were in Table 4. Coupons were sampled over 9 weeks and observed by CLSM microscopy, as described above [27]. For each sample time, five observations were realized. Biovolumes and average thicknesses values were determined with COMSTAT program to compare paints between them [28]. The significance test was conducted using one-way analysis of variance (ANOVA).

### 3.9. Anti-Macrofouling Properties

Paints were applied onto panels (8 × 12 cm). Paints were observed monthly during immersion in natural seawater. The Antifouling performance was assessed according to a modified protocol of the French Standard (NFT34-552 September 1996). Paints were classified using an efficiency parameter N which is expressed as *N* = Σ *I.G* where I stand for the intensity of fouling and G the severity of fouling as shown in Table 5. *N* was determined at each observation time by visual inspection (determination of the surface coverage by each type of fouling) following by a determination of efficiency parameter N referring to Table 5. The lower the *N* value was, the more efficient was the paint.

## Figures and Tables

**Figure 1 ijms-20-00361-f001:**
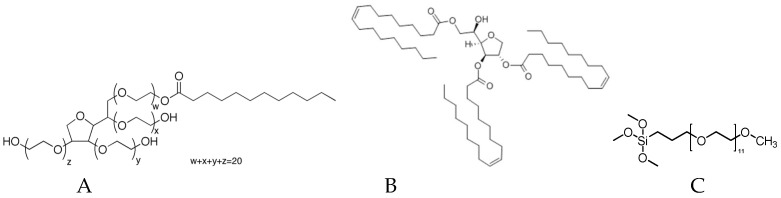
Chemical structure of additives (**A**) Tween 80, (**B**) Span 85, (**C**) PEG-silane.

**Figure 2 ijms-20-00361-f002:**
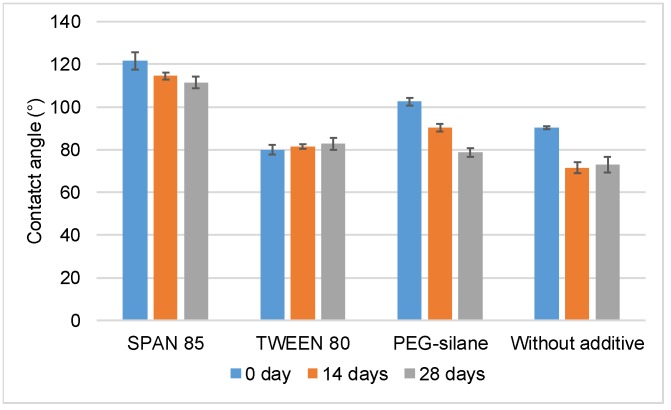
Water contact angles measured on paint surface during their immersion in distilled water.

**Figure 3 ijms-20-00361-f003:**
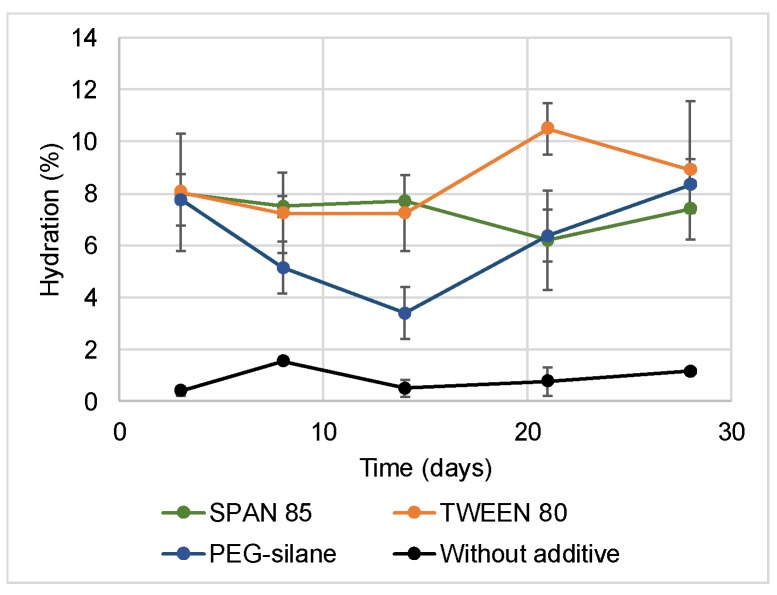
Hydration of paints during immersion measured by Karl Fisher Titration.

**Figure 4 ijms-20-00361-f004:**
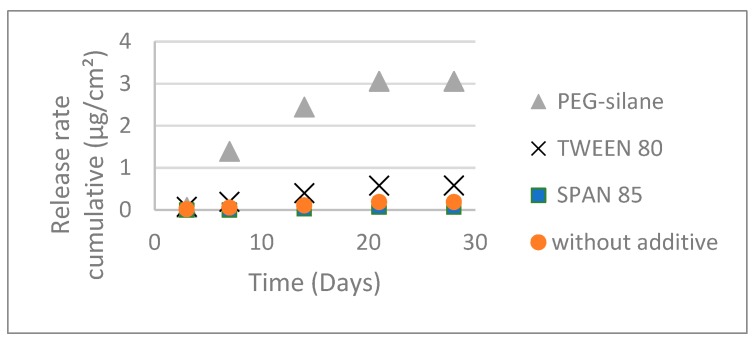
Release of thiocyanate copper from paints during immersion in distilled water.

**Figure 5 ijms-20-00361-f005:**
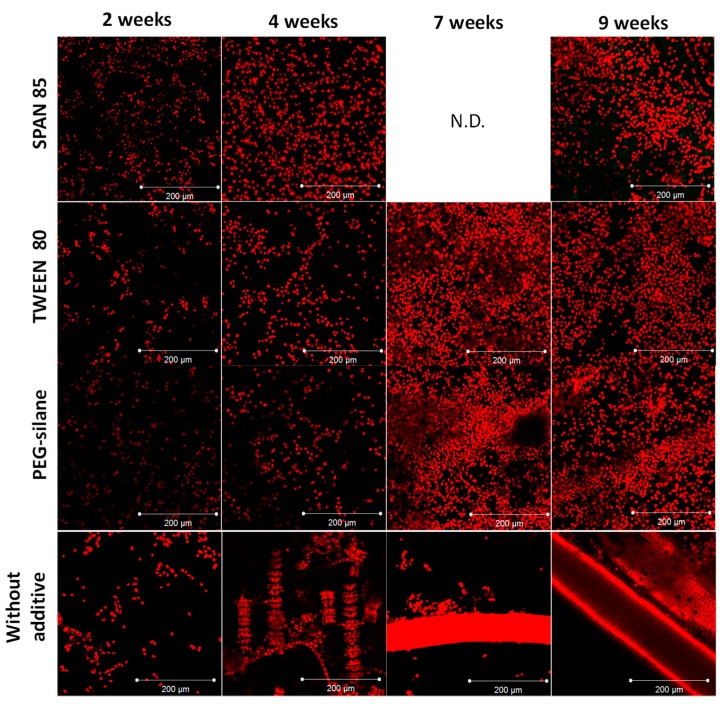
Maximum intensity projection data sets from microalgal biofilms on paints made by confocal laser scanning microscopy after 2, 4, 7 and 9 weeks of immersion in natural seawater (Kernevel Harbour). Microalgae were observed in red by autofluorescence of chlorophyll. N.D. not determined because of technical problems.

**Figure 6 ijms-20-00361-f006:**
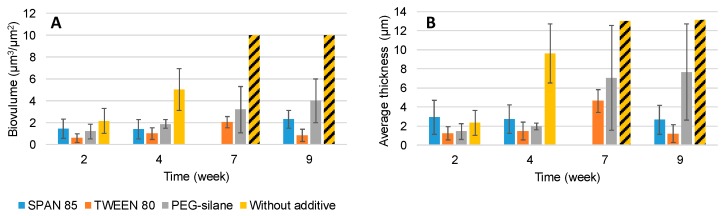
Evolution of A. Biomass and B. Average thickness developed onto paints during immersion. The bare are the mean of five measurements. 
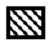
 saturation of biofilm.

**Figure 7 ijms-20-00361-f007:**
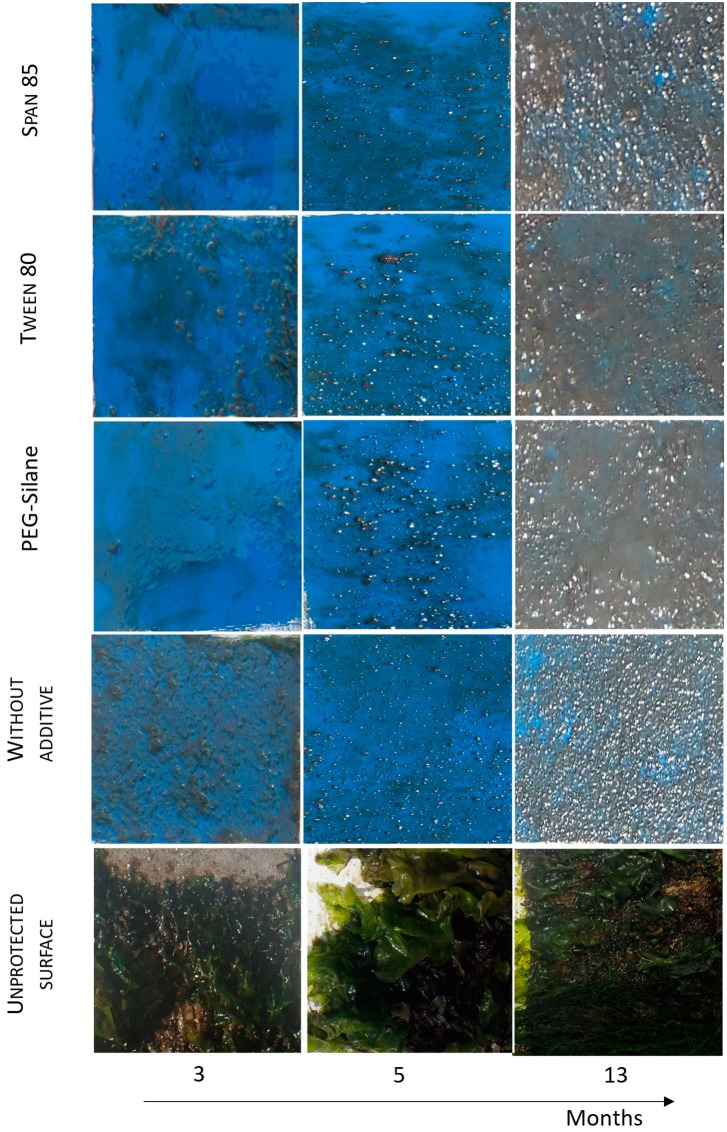
Visual inspection of paints in static condition after 3, 5 and 13 months of immersion in natural seawater.

**Figure 8 ijms-20-00361-f008:**
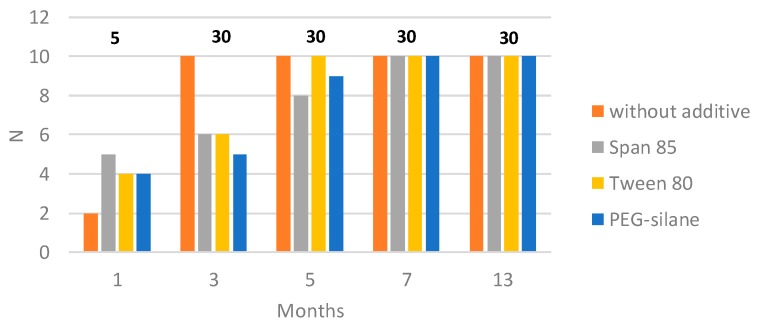
Values of the antifouling efficiency *N* for paints immersed for 1, 3, 5, 7 and 13 months in Atlantic Ocean. Values mentioned on the top of the figure correspond to the *N* values quantified for an unprotected surface.

**Table 1 ijms-20-00361-t001:** Inhibition zone diameter of coatings against *Pseudoalteromonas* sp., *Bacillus* sp., *Paracoccus* sp.

	Inhibition Diameter (cm)		
	*Pseudoalteromonas* sp.	*Bacillus* sp.	*Paracoccus* sp.
Span 85	-	-	-
Tween 80	-	-	0.13 ± 0.09
PEG-silane	-	-	0.07 ± 0.04
Without additive	-	-	-
Commercial paint	0.47 ± 0.11	0.33 ± 0.04	0.37 ± 0.16

- no inhibition diameter was observed.

**Table 2 ijms-20-00361-t002:** Inhibition zone diameter of coatings against *C. closterium, E. gayraliae, P. purpureum*.

	Inhibition Diameter (cm)		
	*Cylindrotheca closterium*	*Exanthemachrysis gayraliae*	*Porphyridium purpureum*
Span 85	0,73 ± 0.15	1.20 ± 0.30	0.60 ± 0.15
Tween 80	0.60 ± 0.10	0.33 ± 0.06	-
PEG-silane	0.46 ± 0.06	0.60 ± 0.06	-
Without additive	-	-	-
Commercial paint	1.2 ± 0.17	2.13 ± 0.15	1.93 ± 0.07

- no inhibition diameter was observed.

**Table 3 ijms-20-00361-t003:** Composition of paints (in wt %).

Product	Composition
Solvents (xylene, methoxypropylacetate)	36
P(CL-VL)	12
Plasticizer	2
CuSCN	20
Additive or CaCO_3_	3
Fillers (wax, silicate, Ti0_2_, ZnO, CaCO_3_)	25
Pigment	2

**Table 4 ijms-20-00361-t004:** Seawater characteristics during immersion of paints.

Month	Temperature (°C)	pH	Conductivity (mS/cm)	Oxygen (mg/L)
1	15.6	8.9	35.7	10.3
3	19.0	7.5	39.4	4.1
6	12.5	8.0	38.1	7.9
13	14	8.5	38.4	10.2

**Table 5 ijms-20-00361-t005:** Determination of efficiency parameter *N*.

Surface Covered by Fouling (%)	Rank for Intensity Factor (I)	Type of Fouling	Rank for Severity Factor (G)
No salissure	0	Biofilm	1
0–10	1	Microalgae	2
10–20	2	Algae (filamentous thallus)	4
20–40	3	Algae (flat thallus)	6
40–60	4	Non-encrusting species	6
60–100	5	Encrusting species	8

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
