# Peer review of "Additives for Efficient Biodegradable Antifouling Paints"

_ijms, 2019, doi:10.3390/ijms20020361_

Reviewer 1 Report

Recommendations to authors:

This manuscript entitled “Additives for biodegradable antifouling paints” described the needs of developing an environmental friendly and new non-toxic anti-fouling system. The authors also pointed out the use of specific additives could improve the efficiency of the paints currently used. Therefore this research evaluated three additives (served as a surface modifier): TWEEN 80, SPAN 85, and PEG-silane. Overall, the topic of this research is interesting and practical, under the perspective of environmental protection especially for marine sciences. Below are recommendations for authors:

1.     Page 4: line 114-116. Figure 4 shows cumulative release of copper during 30 days of immersion. Please clarify the immersion was done in distilled water or seawater? Please state in Figure 4 as well.

2.     Page 5: line 153. In Figure 6, at 7 weeks in both of panel A and Panel B, there were no data points of additive SPAN 85, but data points were recorded at 9 weeks. In addition, both of the biomass and average thickness should increase overtime, however, the 9-week point of TWEEN 80 showed a decreased trend. Please explain.

3.     Page 5: line 158. Please describe how to determine the efficiency factor (N). It is not clear in the text

4.     Page 7: line 180-184. Is the “paint without additive” the same as “commercial paint”? Does the commercial paint contain copper? It is not clear in the text. Please define and use the same terms. (For an example, commercial paint without additive.).

5.     Page 8: line 208-209. “Hence, paints proposed in this paper were composed of……….and kept being efficient.” Please clarify being efficient. Can both of Figure 7 and Figure 8 be used as the supporting data?

Author Response

Page 5: line 153. In Figure 6, at 7 weeks in both of panel A and Panel B, there were no data points of additive SPAN 85, but data points were recorded at 9 weeks. In addition, both of the biomass and average thickness should increase overtime, however, the 9-week point of TWEEN 80 showed a decreased trend. Please explain.

Effectively, no data point of additive SPAN 85 is indicated at 7 weeks because a technical problem has not allow to observe this sample. However, as each data point is realised on independant coupons, it has been make possible to continues the evaluation at 9 weeks.

Biomass and average thickness decrease on paint with Tween80. This observation can be explained by a mechanical erosion of paint surface. This phenomenon has already been observed on erodable paint (Loriot et al., Polymer, 2017, 9, 36 ; Azemar et al, Progress in Organic Coatings, 2015, 87, 10-19)

Page 5: line 158. Please describe how to determine the efficiency factor (N). It is not clear in the text

This part has been improved

Page 7: line 180-184. Is the “paint without additive” the same as “commercial paint”? Does the commercial paint contain copper? It is not clear in the text. Please define and use the same terms. (For an example, commercial paint without additive.).

Other minor point have been taken into consideration

Reviewer 2 Report

The manuscript by Fay et al described the studies about antifouling additives. Authors provided sufficient comparison studies about three modifier (Tween 80, Span 85 and PEG-silane) and their effects for the surface hydration and copper release. Authors provided adequate experiments to support the statement and conclusion. According to the shape of current version, I recommend it published in IJMS as it is. The only part that authors may improve but not necessary is the scale bar is missing on Figure 8.

Author Response

minor point have been taken into consideration

Reviewer 3 Report

Influence of selected additives incorporated in a copper paint on hydrophobicity, hydration and copper release was investigated. The effects of those additives on antifouling efficiency of paints were established. Work was well planned and results of the research were clearly presented. Importantly, obtained results, like anti-bacterial activity, were compared with commercial paint.

Only some minor points require attention:

- line 44 -  “Moreover, copper thiocyanate needs lower amount of biocide than copper oxide for the same efficiency” – copper thiocyanate (which is a biocide) needs lower amount of biocide? -  this sentence should be rewritten.

- Fig. 1 – the quality of part B of this Figure (structure of Span 85) should be improved.

- Fig. 8 – please explain what does the first bar from the left (light blue) represent?

- line 182 -  “No paint affected bacterial growth excepted for Paracoccus sp.” – you mean none of the paints with tested additives? Please clarify this statement.

-line 216 -  could you provide more info about copolymer, like average molecular weight, dispersity index and composition (units ratio).

- line 333 – it should be “wetting” instead of “wettability” in title of publication (according to https://doi.org/10.1016/j.porgcoat.2016.02.018).

- make sure that all DOI numbers are correct, e.g. ref. 25 – there is “20.3390/polym9020036” , while it should be “10.3390/polym9020036”

Author Response

The formulations of paints with and without additive are the same. Only the additive is replaced by calcium carbonate in paint without additive. Commercial paint contains copper and booster biocide and doesn’t contain additive. Hence, commercial paint is well different than paint without additive.

Fig. 8 – please explain what does the first bar from the left (light blue) represent?

The light blue bar is a mistake. The figure has been modified.

“No paint affected bacterial growth excepted for Paracoccus sp.” – you mean none of the paints with tested additives? Please clarify this statement.

The statement has been modified.

line 216 -  could you provide more info about copolymer, like average molecular weight, dispersity index and composition (units ratio).  

Molecular weight and composition have been added in the text.

Other minor point have been taken into consideration